Fish species richness is associated with the availability of landscape components across seasons in the Amazonian floodplain

Carvalho Freitas Carlos Edwar 1 cefreitas@ufam.edu.br
Laurenson Laurie 2
Yamamoto Kedma Cristine 1
Forsberg Bruce Rider 3
Petrere Miguel Jr 4
http://orcid.org/0000-0002-9752-1499 Arantes Caroline 5
Siqueira-Souza Flavia Kelly 1
1 Department of Fisheries Sciences/Faculty of Agriculture Sciences, Federal University of Amazonas , Manaus, Amazonas , Brazil
2 School of Life & Environmental Sciences/Faculty of Science, Engineering & Built Environment, Deakin University , VIC , Australia
3 Coordenação de Dinâmica Ambiental, Instituto Nacional de Pesquisas da Amazonia , Manaus, Amazonas , Brazil
4 Institute of Biological Sciences, Federal University of Pará , Belém, Pará , Brazil
5 Center for Global Change and Earth Observations, University of Michigan—Ann Arbor , MI , USA
Arlinghaus Robert
Electronic publication date: 2018 Jun 21
Publication date: 2018
Volume: 6
Electronic Location ID: e5080
Received 2018 Jan 23; Accepted 2018 Jun 4
Copyright: © 2018 Carvalho Freitas et al.
Copyright year: 2018
Copyright holder: Carvalho Freitas et al.
License: This is an open access article distributed under the terms of the Creative Commons Attribution License, which permits unrestricted use, distribution, reproduction and adaptation in any medium and for any purpose provided that it is properly attributed. For attribution, the original author(s), title, publication source (PeerJ) and either DOI or URL of the article must be cited.
License URL: https://creativecommons.org/licenses/by/4.0/

Keywords: Fish diversity, Spatial environmental variables, Wetlands, Amazon basin

Funding: CNPq 302807/2015-2 FINEP (PIATAM Project) CNPq 200893/2012-2 Applied Biodiversity Science Program (ABS/NSF-IGERT) Texas A&M University Carlos Edwar Carvalho Freitas was funded by CNPq (grant 302807/2015-2) and FINEP (PIATAM Project). Caroline Arantes was funded by CNPq (200893/2012-2), Applied Biodiversity Science Program (ABS/NSF-IGERT), Tom Slick Fellowship and Dissertation Fellowship—Texas A&M University. The funders had no role in study design, data collection and analysis, decision to publish, or preparation of the manuscript.

==============================
Understanding environmental biodiversity drivers in freshwater systems continues to be a fundamental challenge in studies of their fish assemblages. The present study seeks to determine the degree to which landscape variables of Amazonian floodplain lakes influences fish assemblages in these environments. Fish species richness was estimated in 15 Amazonian floodplain lakes during the high and low-water phases and correlated with the areas of four inundated wetland classes: (i) open water, (ii) flooded herbaceous, (iii) flooded shrubs and (iv) flooded forest estimated in different radius circular areas around each sampling site. Data were analyzed using generalized linear models with fish species richness, total and guilds as the dependent variable and estimates of buffered landscape areas as explanatory variables. Our analysis identified the significance of landscape variables in determining the diversity of fish assemblages in Amazonian floodplain lakes. Spatial scale was also identified as a significant determinant of fish diversity as landscape effects were more evident at larger spatial scales. In particular, (1) total species richness was more sensitive to variations in the landscape areas than number of species within guilds and (2) the spatial extent of the wetland class of shrubs was consistently the more influential on fish species diversity.

Introduction

Floodplains are key environments for the health of large river ecosystems (Junk et al., 2014), as they regulate water flow and nutrients that are essential for the life cycle of many species (Junk, Bayley & Sparks, 1989; Fernandes, 1997). Floodplains of the Amazon River are highly productive, with estimations of total net primary productivity reaching 300 Tg C year−1 in an area of 1.77 × 106 km2 (Melack et al., 2009). The high productivity of the Amazonian floodplain is driven in part by seasonal changes in water level that can exceed 15 m and lead to remarkable spatio-temporal changes in the landscape (i.e., flood pulse, Junk, Bayley & Sparks, 1989). This high spatio-temporal heterogeneity of habitats across the landscape, including open water, macrophyte meadows, flooded shrubs, forests and herbaceous regions, and large extensions of ecotones integrated by a complex chain of connections, influence fishes movement, feeding and reproductive behaviors as well as their growth and survival rates (Petry, Bayley & Markle, 2003; Siqueira-Souza & Freitas, 2004; Freitas et al., 2010a; Siqueira-Souza et al., 2016).

While influences of local habitat features in these ecosystems on fishes have been relatively well studied (e.g., structural or physic-chemical variables measured within lakes) (e.g., Rodriguez & Lewis, 1997; Tejerina-Garro, Réjean & Rodriguez, 1998; Miranda, 2011; Freitas et al., 2014), influences of landscape features have been generally overlooked. The few studies evaluating this issue have found landscape components, such as habitat heterogeneity (e.g., number or size of habitats) and land cover types (e.g., forest cover) influence both fish diversity and fish biomass (Yager, Layman & Allgeier, 2011; Siqueira-Souza et al., 2016; Lobón-Cervia et al., 2015; Arantes et al., 2017; Castello et al., 2017). Given the magnitude of the Amazonian floodplain, its variety of habitats (Siqueira-Souza et al., 2016; Freitas et al., 2010b; Hurd et al., 2016) and increasingly anthropogenic-driven impacts on its landscapes, a continuing understanding of this issue is critically needed. Particularly, there is a need to understand how fish diversity is influenced by the flood-pulse driven availability of habitat within the landscape.

Herein, we tested the hypothesis that landscape components, represented by the spatial extent of four landscape variables (open water, flooded herbaceous, flooded shrub and flooded forest), significantly influenced fish diversity in Amazonian floodplain lakes, with the importance of each landscape variable being dependent on the hydrological period (i.e., high- and low-water). Specifically, we evaluated how taxonomic and functional species richness, measured in high- and low-water periods, responded to the availability of landscape features that surrounded the sampled lakes. This evaluation provided an understanding of the regional species pool and the importance of landscape spatial scales to the fish assemblages and has significant usefulness for the long-term conservation of Amazonian floodplains (Freitas et al., 2014).

Materials and Methods

Study area

Fish assemblages were sampled in 15 lakes on the central Amazon floodplain along the middle and lower Solimões River, 11 located on the margins of the main channel and four within fluvial islands. The Solimões is a whitewater river (the color of heavily creamed coffee) as consequence of the high load of suspended nutrients (Furch, 1984). It flows through a geologically recent system (Hoorn et al., 2010) and is geomorphologically monotonous in its physical structure (Latrubesse & Franzinelli, 2002). The associated floodplain is an active alluvial system still at work, blanketing and reworking the floodplain deposits (Latrubesse & Franzinelli, 2002). All 15 sampled lakes are located within this floodplain and are in general shallow lakes formed behind scroll bars by the overbank deposition of fine material.

The spatial extent of landscape features is substantially different between high and low-water periods (Figs. 1 and 2). The central Amazon floodplain is a dynamic sedimentary formation that includes both marginal plains and isolated islands that are continuously re-worked by fluvial erosion and sedimentation (Dunne et al., 1998). The geomorphological processes result in variations in elevation and inundation, which have a fundamental effect on the distribution and dynamics of the floodplain vegetation and habitats (Junk, Bayley & Sparks, 1989; Schöngart et al., 2002). As the elevation declines and inundation period increases, floodplain vegetation and habitats change from alluvial forest to shrubs, herbaceous vegetation and finally, to open water. These habitats are only available to fish when flooded and the extent of flooded habitats may vary by a few orders of magnitude as the Amazon main channel undergoes its annual 10–12 m flood cycle (Hess et al., 2003).

Figure 1 Map of study area at the lower stretch of the Solimões River showing the lakes sampled during high water season and the buffers of 500, 1,000 and 5,000 m.

At this season, the aquatic environments are expanded and the landscape is mostly aquatic and lakes, river and channels are connected.

Figure 2 Map of the study area showing the lakes sampled during low water seasons and the buffers of 500, 1,000 and 5,000 m.

Landscape component estimates

Hess et al. (2003, 2015) published a dual-season (high and low-water) wetland classification for the central Amazonian region, derived from 100 m L-band synthetic aperture radar imagery. Nine landscape classes were identified for high and low-water conditions. Four landscape variables defined by Hess et al. (2003) were used in our analytical models: Open Water—also called non-vegetated flooded, represents aquatic habitats without vegetation cover and includes lakes and secondary channels; Flooded Herbaceous—vegetation dominated by non-woody plants, with <25% trees or shrubs, the herbaceous cover is usually ≥25% but may be less if it exceeds that of other vegetation; Flooded Shrubs—vegetation dominated by low stature (height 0.5–5 m) woody plants, with individuals or clumps not touching or interlocking, shrub cover is usually ≥25% but may be less if it exceeds that of other vegetation; and Flooded Forest—closed canopy forest dominated by woody plants >5 m in height, with interlocking crowns, generally forming 60–100% of crown cover. The thematic wetland maps for high and low-water periods were downloaded from NASA’s Oak Ridge National Laboratory Distributed Active Archive Center (https://daac.ornl.gov/cgi-bin/dsviewer.pl?ds_id=1284) and imported into ArcMap 10.1, together with fish sampling points digitized from GPS coordinates. Using the Spatial Analyst—Extract by Circle Tool, we quantified the areas of landscape variables (m2) in 500, 1,000 and 5,000 m (radius) circular buffers around each fish sampling site during low and high-water periods (Figs. 1 and 2). The buffer radii were chosen to allow the characterization of landscape features at different spatial scales: (i) 500 m radius represents the area immediately surrounding the sampling sites, it remains completely inundated for most of the year; (ii) 1,000 m radius represents the area surrounding the sampling sites and includes some areas inundated during the flood season; and, (iii) 5,000 m represents the area surrounding the sampling sites plus areas that are inundated only during the high-water season and includes secondary channels, other lakes and the main river channel. Further, we adapted the approach employed by Lobón-Cerviá et al. (2015) who defined buffers based on the swimming ability of a hypothetic fish with pre-defined body size and swimming speed. Our analyses considered that fishes have diverse swimming and dispersal abilities. Thus, the definition of the buffers seeks to delimit areas that are potentially explored by these fishes according to their different ecological strategies, including their different migratory behaviors.

Fish samplings

Fish assemblages were sampled using 11 standardized floating gillnets 15 m long and 2 m high with varying stretched mesh sizes (30, 40, 50, 60, 70, 80, 90, 100, 110, 120 and 130 mm). Gillnets were deployed across all representative habitats in each lake system. Although floating gillnets show selectivity towards pelagic and benthopelagic species, they are easier to standardize than other fishing gear and were thus chosen. Furthermore, we note that the várzea lakes are shallow which allows the nets to normally fish across the majority or all of the water column (depending on the site). Nets were set at 06:00 am and remained in the water for 12 h in Ananá, Araça, Baixio, Iauara, Maracá, Poraqué, and Preto lakes; and for 48 h in Cacauzinho, Calado, Camaleão, Camboa, Central, Padre, Santo Antonio and Sacambu lakes. The differences in the duration of fishing times resulted from the inclusion of data from multiple independent research projects. However, sampling effort can be standardized and in the present study it was defined as the product of the number of samplings and sampling time (hours; Supplemental Information 1) and the efficacy of the fish assemblages sampling was evaluated using rarefaction curves (Supplemental Information 4). Gillnets were inspected every 6 h to minimize predation on captured fishes. Sampled fishes were euthanized by thermal shock and were usually identified in the field. Unidentified specimens were fixed in 10% formalin and identified later in the laboratory. While sampling frequency varied among lakes, each lake was sampled at least once during both high and low-water periods (Supplemental Information 1). Fish samplings were done under licenses 30052-1, 50662-1 (Instituto Chico Mendes de Conservação da Biodiversidade—ICMBio/Brazil).

Data analysis

Generalized Linear Models (GLM) based on a Poisson distribution of probability were used to evaluate relationships between fish assemblages and landscape area for each buffer for high and low-water periods. We first modeled total species richness as response variables and the areas of Open Water, Flooded Herbaceous, Flooded Shrub and Flooded Forest as explanatory variables. Then, we classified species according to their trophic guilds, and modeled the richness of carnivorous, omnivorous, and herbivorous species as response variables, and landscape areas for each buffer as independent variables. Also, we classified species according to their migratory behavior and modeled the richness of migrant and resident species as response variables and the same pool of landscape areas as independent variables. And finally, we classified species by the preferential position at the water column as pelagic and benthopelagic to run similar analytical models. Because the number of fish sampled can be correlated with species richness (Angermeier & Schlosser, 1989), we included in the model fish abundance as an independent variable. Models fitted to the low-water data were constrained to the 5,000 m buffer as Flooded Shrubs were absent in both, 500 and 1,000 m buffers. Scatter-plots with trend-lines were presented for variables with statistically significant relations. Landscape spatial scales (i.e., different buffer sizes and associate landscape attributes) were compared at high and low-water using the explained deviance (pseudo-R2), which was also used to assess model fit.

To minimize the effects of auto-correlation between landscape variables (Dormann et al., 2013), an aggregate variance inflation factor (VIF) smaller than 2 was used as a criterion for deciding whether particular variables were included in the models. The Flooded Forest variable showed strong collinearity in most models and was only included in the high-water model for the 5,000 m buffer (Supplemental Information 2). As collinearity between environmental variables was not constant in space, we also used Moran’s I statistic (Fortin, Drapeau & Legendre, 1989) to test for spatial auto-correlation in model residuals. Models for richness of carnivorous and omnivores species as function of the landscape variables in the 1,000 m buffer showed significative spatial autocorrelation; therefore, were not presented (Supplemental Information 2).

Model fits were assessed by visual inspections of the residuals and only those that did not violate the assumptions of the generalized linear models were considered. The Bonferroni correction was employed to adjust for the effect of multiple statistical tests performed on the significance of explanatory variables (Shaffer, 1991).

All statistical analyses were conducted using R Statistical Software (R Core Team, 2013). GLM were fitted using the MASS Package (Ripley et al., 2013). VIF were estimated using the CAR Package (Fox & Weisberg, 2011) and Moran’s I estimates were calculated using the APE Package (Paradis, Claude & Strimmer, 2004).

Results

A total of 178 species was collected. Characiformes was the most diverse group with 73 species, followed by Siluriformes with 64 species. A total of 32 and 52 species were present in more than 75 and 50% of lakes, respectively (Supplemental Information 3). Species richness varied between 52 in Santo Antonio Lake and 89 in Maracá Lake. Species richness was consistently higher during low than in high-water periods. Trophic guilds richness was generally constant among lakes in high and low-water seasons, but the number of resident species was generally higher than the number of migrant species in all lakes (Table 1).

Table 1 Ecological measures for each lake and hydrological season.

Lake	S	SC	SO	SH	M	R	P	B	
HW	LW	HW	LW	HW	LW	HW	LW	HW	LW	HW	LW	HW	LW	HW	LW	
Ananá	65	74	18	21	18	17	65	74	24	26	31	32	22	23	43	50	
Araçá	63	73	16	16	18	18	63	73	20	21	29	35	17	21	46	52	
Baixio	59	75	18	19	17	17	59	75	20	20	26	34	18	19	39	56	
Cacauzinho	60	62	14	14	16	17	60	62	20	21	23	26	17	18	43	44	
Calado	63	62	15	14	17	17	63	62	19	18	23	24	22	21	40	40	
Camaleão	79	57	21	20	19	18	79	57	23	20	32	27	23	20	55	36	
Camboa	66	72	18	18	16	16	66	72	23	24	31	34	23	25	43	47	
Central	57	65	11	13	14	15	57	65	23	23	25	28	22	22	35	43	
Iauara	59	80	16	16	17	17	59	80	18	18	26	36	17	22	42	58	
Maracá	89	62	22	21	29	20	89	62	29	28	33	29	26	17	62	45	
Padre	54	82	12	12	11	12	54	82	23	22	25	34	14	27	40	55	
Poraqué	58	55	10	10	11	11	58	55	23	23	25	26	21	12	37	43	
Preto	62	55	13	13	19	18	62	55	23	21	22	17	16	15	46	40	
Sacambú	61	74	11	12	15	16	61	74	24	25	27	36	20	28	41	46	
Santo Antonio	52	69	14	14	11	12	52	69	29	20	25	33	22	24	30	43	
Note:

HW, high-water; LW, low-water (LW); S, species richness; SC, carnivorous richness; SO, omnivorous richness; SH, herbivorous richness; M, migrant richness; R, resident richness; P, pelagic richness; B, benthopelagic richness.

During the high-water season, flooded shrub area was the only significant landscape variable (Table 2). Total species richness tended to be greater where shrub cover was greater in different buffer scales (500 m and 1,000 m) (Table 2; Figs. 3A and 3B). This pattern also was observed for resident species (Table 2; Figs. 3C and 3D). Omnivorous species richness was, again, positively related with shrub cover but just in the small-scale (buffer of 500 m) (Table 2; Fig. 3E).

Table 2 Summary of general linear models.

Model	df	OW	FH	FS	FF	N	Pseudo-R2	
Total species richness	
500 m/high water	11	Ns	Ns	0.02*	ex	Ns	0.47	
500 m/low water	11	Ns	0.07#	Ni	Ns	Ns	0.39	
1,000 m/high water	12	Ns	Ns	0.01*	ex	Ns	0.47	
1,000 m/low water	11	0.001*	0.02*	Ni	ex	Ns	0.56	
5,000 m/high water	11	Ns	Ns	Ns	ex	Ns	0.37	
5,000 m/low water	11	Ns	0.003*	Ns	ex	0.001*	0.52	
Carnivorous richness	
500 m/high water	11	Ns	Ns	Ns	ex	Ns	0.35	
500 m/low water	12	Ns	Ns	Ni	−0.018#	Ns	0.46	
1,000 m/high water	11	Ns	Ns	Ns	ex	Ns	0.36	
1,000 m/low water	12	Ns	Ns	Ni	ex	Ns	0.18	
5,000 m/high water	11	Ns	Ns	Ns	ex	Ns	0.42	
5,000 m/low water	11	Ns	Ns	Ns	ex	Ns	0.32	
Omnivorous richness	
500 m/high water	10	Ns	Ns	0.04*	Ns	Ns	0.46	
500 m/low water	11	Ns	Ns	Ni	Ns	Ns	0.20	
1,000 m/high water	11	Ns	Ns	0.01*	ex	Ns	0.48	
1,000 m/low water	12	Ns	Ns	Ni	ex	Ns	0.12	
5,000 m/high water	11	Ns	Ns	Ns	ex	Ns	0.18	
5,000 m/low water	11	Ns	Ns	0.001#	ex	Ns	0.34	
Herbivorous richness	
500 m/high water	11	Ns	Ns	Ns	ex	Ns	0.59	
500 m/low water	11	Ns	Ns	Ni	Ns	Ns	0.07	
1,000 m/high water	11	Ns	Ns	Ns	ex	Ns	0.35	
1,000 m/low water	12	Ns	Ns	Ni	ex	Ns	0.02	
5,000 m/high water	11	Ns	Ns	Ns	ex	Ns	0.34	
5,000 m/low water	11	Ns	Ns	Ns	ex	Ns	0.50	
Migrant richness	
500 m/high water	11	Ns	Ns	Ns	ex	Ns	0.33	
500 m/low water	11	Ns	Ns	Ni	Ns	Ns	0.15	
1,000 m/high water	11	Ns	Ns	Ns	ex	Ns	0.40	
1,000 m/low water	12	Ns	Ns	Ni	ex	Ns	0.23	
5,000 m/high water	11	Ns	Ns	Ns	ex	Ns	0.55	
5,000 m/low water	11	Ns	Ns	Ns	ex	Ns	0.22	
Resident richness	
500 m/high water	11	Ns	Ns	0.03*	ex	Ns	0.39	
500 m/low water	12	0.04#	Ns	Ni	ex	Ns	0.56	
1,000 m/high water	11	Ns	Ns	0.01*	ex	Ns	0.34	
1,000 m/low water	12	0.01*	Ns	Ni	ex	Ns	0.66	
5,000 m/high water	11	Ns	Ns	Ns	ex	Ns	0.19	
5,000 m/low water	11	Ns	Ns	Ns	ex	Ns	0.35	
Pelagic richness	
500 m/high water	11	Ns	Ns	Ns	ex	Ns	0.53	
500 m/low water	11	Ns	Ns	Ni	Ns	Ns	0.06	
1,000 m/high water	11	Ns	Ns	Ns	ex	Ns	0.47	
1,000 m/low water	12	Ns	Ns	Ni	ex	Ns	0.24	
5,000 m/high water	10	Ns	Ns	Ns	Ns	Ns	0.18	
5,000 m/low water	11	Ns	0.001#	Ns	Ex	Ns	0.33	
Benthopelagic richness	
500 m/high water	11	Ns	Ns	0.02#	ex	Ns	0.45	
500 m/low water	12	Ns	0.09#	Ni	ex	Ns	0.47	
1,000 m/low water	12	Ns	Ns	Ni	ex	Ns	0.41	
5,000 m/high water	11	Ns	Ns	Ns	ex	Ns	0.39	
5,000 m/low water	11	Ns	Ns	Ns	ex	Ns	0.34	
Notes:

Species richness is the response variable and open water area (OW), flooded herbaceous area (FH), flooded shrubs area (FS), flooded forest area (FF) and number of sampled fish (n) were explanatory variables. Model coefficients are exhibited when they are significant at least for p < 0.10.

Obs: df, residual degrees of freedom; ex, previously excluded by collinearity; Ni, not included in the model; Ns, not significant.

# 0.10 < p < 0.05.

* 0.05 < p < 0.01.

Figure 3 Scatter-plots by buffer and high-water season.

At 500 m—(A) total species richness vs. flooded shrub; (B) omnivorous richness vs. flooded shrub; (C) resident richness vs. flooded shrub; 1,000 m—(D) Total sp vs. flooded and (E) resident richness vs. flooded shrub. Tendency lines for the models with significative independent variables, where points are observed values of species richness per site, full lines are fitted lines and dotted lines are confidence intervals at 95%.

During the low-water season, flooded herbaceous had the strongest relationship with fish diversity (Table 2). Flooded herbaceous was positively related to total species richness for the larger (5,000 m) and medium (1,000 m) buffers (Table 2; Figs. 4A and 4B). A second influential landscape variable was open water that positively affected total species richness and resident species richness (Table 2; Figs. 4C and 4D). The number of fish sampled positively influenced total species richness at the largest buffer, and showed a slightly positive, but not statistically significant, relation with carnivorous richness (p < 0.10) (Table 2; Fig. 4E).

Figure 4 Scatter-plots by buffer during low-water season.

At 1,000 m—(A) total species richness vs. open water, (B) resident richness vs. open water, and (C) total species richness vs. flooded herbaceous; and 5,000 m—(E) total species richness vs. flooded herbaceous and (F) total species richness vs. number of individuals. Tendency lines for the models with significative independent variables, where points are observed values of species richness per site, full lines are fitted lines and dotted lines are confidence intervals at 95%.

The best models with respect to the scale of analysis varied depending on the season and fish group analyzed. During high-water, the small and medium sized buffers (500 m and 1,000 m buffers, respectively) explained more variability in the relationship between total species richness and landscape variables (pseudo-R2 > 0.4, Table 2). During this season, models with omnivorous and herbivorous species richness as response variables, showed better fits for the small (500 m) buffer (pseudo-R2 > 0.45) than larger buffer sizes, although the results for herbivores was not significant at any scale. Models for carnivorous and migrant species richness during high-water showed weak relationships with the landscape variables at all scales of analysis (pseudo-R2 < 0.40, Table 2).

During low-water, best fits were observed for the 500 and 1,000 m buffers (Table 2). During this season, the best-fitted model was observed for the medium buffer when using resident species as the response variable (1,000 m buffer, pseudo-R2 = 0.66).

Discussion

Our results demonstrated that fish assemblages in the Amazon basin respond to changes in the spatial extent of landscape components and that these responses vary depending on the ecological strategies of the fish and on the stage of the hydrological cycle. It is evident that fish species richness is related to the extent of shrub vegetation found in surrounding floodplain lakes during high-water, and to the extent of herbaceous and open water regions during low-water periods. Within this framework though, omnivorous and resident species, regardless of season, showed stronger correlations with these categories than carnivorous and migratory species. The results support the view that degradation of Amazonian floodplain landscapes can impact fish diversity (Lobón-Cervia et al., 2015), with certain groups being more vulnerable than others (Arantes et al., 2017).

Furthermore, our results support the contention that seasonal variations in landscape components impact different groups of fishes at different times, and thus drive cross-habitat migration within floodplains driven by flood-pulses (Fernandes, 1997; Castello, 2008). The observed rise in resident species (e.g., Osteoglossum bicirrhosum, Cichla monoculus) richness with increasing shrub cover, could reflect the lateral migration of these species into these flooded habitats. It is likely that the flooded shrubs represent suitable regions for reproduction, feeding and/or refuge. An observation that is also true for omnivorous species richness (e.g., Triportheus spp). It is also possible that there is a direct association between the spatial extent of shrub cover and the extent of flooding during the high-water periods, that is, the observed extent of this habitat class is maintained by the annual flooding regime (Silva, Costa & Melack, 2010; Hess et al., 2015). Should this be true, then we can hypothesize that the regional abundance of shrubs together with their structural complexity are responsible for seasonality in fish diversity. Studies have demonstrated that during low-water, many fish species migrate from flooded habitats to lakes and secondary channels (Fernandes, 1997). Our observation that during this season species richness is related to the amount of open water, is consistent with previous studies showing that fish seek out lakes with greater water volumes (e.g., deep lakes) possibly to avoid the effects of extreme droughts (Arantes et al., 2013).

A strong positive correlation is often expected between sample number and species richness (Angermeier & Schlosser, 1989). However, in our results the number of individuals was only related to species richness in the low-water model. It is possible that this result is derived from high populational densities and consequently higher catch rates of our sampling gears during these periods of water retraction.

Our results showing differential influences of landscape components on fish diversity across scales are consistent with previous studies in the Amazon floodplain areas (Freitas et al., 2014; Siqueira-Souza et al., 2016) and support the view that spatial scale of investigation on the processes affecting biodiversity matters (Chase & Leibold, 2002; Willis & Whitaker, 2002; Rahbek, 2004). The medium scale buffer (500 m) explained high variability in the relationships between fish diversity and landscape components for both seasons (low and high-water) and, therefore, may represent an appropriate scale of analysis. However, further understanding of the scales of processes driving spatial fish diversity patterns could be achieved by exploring other scales, including different buffer sizes along with a local catchment (i.e., lake system sensu Arantes et al., 2017; Castello et al., 2017).

The large amount of variation that remained unexplained by the landscape variables in our results is likely related to the high spatial-temporal heterogeneity and variability of these floodplain habitats and associated local environmental variables (Freitas et al., 2014; Siqueira-Souza et al., 2016; Junk, Bayley & Sparks, 1989; Röpke et al., 2016; Hurd et al., 2016). The large dimensions and heterogeneity of the Amazon floodplain coupled with strong temporal variations result in a complex ecosystem whose structure and dynamics are governed by deterministic and stochastic mechanisms operating across a broad range of temporal and spatial scales (Freitas et al., 2014; Hess et al., 2015). Yet a large set of variables including several that we did not include in our models, have been found to influence populations and assemblage dynamics in these floodplains (e.g., depth, transparency, dissolved oxygen, connectivity, Rodriguez & Lewis, 1997; Freitas et al., 2014; Kemenes & Forsberg, 2014; Siqueira-Souza et al., 2016; Hurd et al., 2016; Lobón-Cervia et al., 2015). Habitat variations between high- and low-water may, therefore, represent only part of the variations affecting fish community structure in this system. Including these variables along with landcover components may reveal stronger spatial patterns of fish diversity across the hydrological cycle.

Understanding the importance of landscape components in structuring freshwater fish assemblages is a core question in biological conservation, especially in light of the increasing loss of aquatic habitats and the threat it poses to freshwater fish diversity globally (Dudgeon et al., 2006). Our results demonstrate that fish species richness is closely linked to habitat composition at the landscape scale and suggest that increasing losses of aquatic habitats in the Amazon due to deforestation and river impoundment (Kahn, Freitas & Petrere, 2014; Lees et al., 2016; Lobón-Cervia et al., 2015; Arantes et al., 2017; Castello et al., 2017; Forsberg et al., 2017) could disrupt these complex relationships with unpredictable consequences on Amazonian fish diversity.

Conclusions

We conclude that the total species richness was more sensitive to variations in the landscape areas than number of species within guilds and the spatial extent of the wetland class of shrubs was consistently the more influential on fish species diversity. In synthesis, our results highlight the importance of the geospatial extent of landscape variables surrounding Amazonian lakes systems in the maintenance of their fish diversity.

Supplemental Information

Supplemental Information 1 Studied lakes: their geographical coordinates, number of samplings in each phase of the hydrological cycle (HW = high water, LW = low water), sampling effort (in hours of the net in the water) and area (km2).

Click here for additional data file.

Supplemental Information 2 Summary of assumptions tested by each model.

Total S = total species richness, Carnivorous S = carnivorous species richness, Omnivorous S = omnivorous species richness, Herbivorous S = herbivorous species richness, Migrant S = migrant species richness, Resident S = resident species richness, Pelagic P = pelagic species richness, Benthopelagic B = benthopelagic species richness, OW = open water, FH = flooded herbaceous, FS = flooded shrubs, FF = flooded forest, n = number of collected fish.

Click here for additional data file.

Supplemental Information 3 List of species by lake and season of the hydrological cycle (H = high water and L = low water).

Where x = present and … = absent. And type of functional group as C = carnivorous, I = omnivororus, H = herbivorous, R = resident, M = migratory, P = pelagic and B = benthopelagic.

Click here for additional data file.

Supplemental Information 4 Rarefaction curves, where H = high-water and L = low-water.

Click here for additional data file.

Additional Information and Declarations

Competing Interests

Author Contributions

Animal Ethics

Data Availability

The authors declare that they have no competing interests.

Carlos Edwar Carvalho Freitas conceived and designed the experiments, analyzed the data, prepared figures and/or tables, authored or reviewed drafts of the paper, approved the final draft.

Laurie Laurenson conceived and designed the experiments, analyzed the data, authored or reviewed drafts of the paper, approved the final draft.

Kedma Cristine Yamamoto conceived and designed the experiments, performed the experiments, authored or reviewed drafts of the paper, approved the final draft.

Bruce Rider Forsberg analyzed the data, authored or reviewed drafts of the paper, approved the final draft.

Miguel Petrere Jr authored or reviewed drafts of the paper, approved the final draft.

Caroline Arantes analyzed the data, prepared figures and/or tables, authored or reviewed drafts of the paper, approved the final draft.

Flavia Kelly Siqueira-Souza conceived and designed the experiments, performed the experiments, prepared figures and/or tables, authored or reviewed drafts of the paper, approved the final draft.

The following information was supplied relating to ethical approvals (i.e., approving body and any reference numbers):

Instituto Chico Mendes de Conservação da Biodiversidade—ICMBio provided full approval for fish samplings (licenses: 30052-1 and 50662-1).

The following information was supplied regarding data availability:

The raw data are provided in the Supplemental Files.

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
