# Peer review of "Fish species richness is associated with the availability of landscape components across seasons in the Amazonian floodplain"

_PeerJ, doi:10.7717/peerj.5080_

## Round 0.1 · original submission · Major Revisions

I agree with the comments by the reviewers, which mainly deal with lack of clarity in some sampling aspects and site descriptions and the statistical analysis. To reiterate two important points that I think the revision should attempt. First, it would be nice if you could justify the choice of your landscape variables and in particular their dimensions better. What is the reason to expect these dimensions having effects, why not, say, 550 m? Second, I am concerned about the statistical analysis. You essentially provided many independent biviriate models without correcting the p values for multiple comparisons. This can led to biased inferences. I strongly suggest you search for an elegant multivariate analysis technique common in community ecology both in relation to the dependent and to the independent variables and attempt to use dimension reduction techniques.

·

Basic reporting

no comment

Experimental design

no comment

Validity of the findings

No comment

Additional comments

Dear editor

I reviewed the manuscript "Landscape components and flood pulses drive fish species richness in Amazonian floodplain lakes", which brings evaluations among how taxonomic and functional species richness, measured in high and low-water periods in Amazon lakes. The authors suggest that the total species richness was more sensitive to variations in the landscape areas than number of species within guilds and the spatial extent of the wetland class of shrubs was consistently the more influential on fish species diversity. Authors provided careful samplings and a satisfactory analytical protocol. I have some minor comments that I believe that authors should discuss in their manuscript.

Specific comments and suggestions:

Line #88
Inform the size of lakes

Line #114
"…we quantified the areas of
landscape variables (m2) in 500, 1,000 and 5,000 m (radius)… "

Explain how the size of the buffers (500, 1,000 and 5,000 m) has been determined and whether there is any relation to the size of the ponds.

Not a question that would justify the use of different buffers is shown.

Line #118
“…11 standardized floating gillnets 15 m long and 2 m deep with
varying stretched mesh sizes (30, 40, 50, 60, 70, 80, 90, 100, 110, 120 and 130 mm).”

Comment on whether these nets are floating or at the bottom of the river. This may reflect on sampling and specie group colleted.

Line #120
“ … remained in the water for 12 hours in Ananá, Araça, Baixio, Iauara, Maracá, Poraqué, and Preto lakes; and for 48 hours in Cacauzinho, Calado, Camaleão, Camboa, Central, Padre, Santo Antonio and Sacambu lakes.”

There is some reason for the differences in the time of exposure of nets in the lakes, How this was controlled in the analyzes.

Line #122
“Sampling effort among lakes were standardized as the product of the number of samplings and sampling time (hours)”

You need a better explanation. Is this a CPUE? It is confusing because it is mentioned that only the richness of species was used.

Line #126
“While sampling frequency varied among lakes, each lake was sampled at least once during both high and low water
periods (Fig. S1).”

What was the effort for each point in the two seasons? And how this effect was controlled.

Line #132
“…buffer for high and low-water periods. We first modeled total species richness as response variables and the are…”

What is the reason to use only the number of species and not the species composition?

Line #139
“…we included in the model species abundance as an independent variable. Models fitted to the low-water data were constrained to the…”

I believe you need to make clear what is being said about species richness, in here you used abundance.

Line #147
How it was worked with the autocorrelation that occurs between the different sizes of buffers since the buffer of 500 is inside the 1000 and this one is inside 5000.

Line#160
“A total of 178 species was collected.”

I suggest to present in supplementary material the list of species, trophic category, migration, and presence/absence by lake.

Line#169

“Omnivores species richness was, again, positively related with shrub cover but just in the small-scale (buffer of 500m) (Table 2, Fig. 3e).”

This information had not been mentioned yet.

Line #199
“… flood-pulses (Fernandes, 1997; Castello, 2008). The observed rise in resident species (e.g. Osteoglossum bicirrhosum, Cichla monoculus)…”

I would suggest a list of species in Supplementary Material.

Line #212
"The significance of flooded herbaceous areas as refugia for several prey species during the high-water could not be clearly quantified as several groups of herbivores, carnivores, and migratory species did not have any significant relationship with the extent of this landscape variable. We note that herbivorous fishes, and possible migratory species, have previously been shown to have strong relationships with forest cover (Arantes et al., 2017), but we were not able to assess this."

Paragraph loose in the text.

·

Basic reporting

Dear author of the manuscript “Landscape components and flood pulses drive fish species richness in Amazonian floodplain lakes”,
I really like your research question and I think these data can result in a good paper. Furthermore I like your way of writing, as it is mostly clear and easy to understand. Only in one section of the MATERIALS AND METHODS some improvement is needed to clarify the methods.
Line 113-115: Could you please mention that the landscape variables in your circular buffer are the four landscape variables defined by Hess et al. (2003). I find this section confusing.

In the tables and figure I found some mistakes, but as I also recommend a new data analyses, the tables and figures need to be renewed anyway.
Table 1: Santo Antonio: 1229?
Table 2: -0.018#?
Figure 3: figure fuzzy; why do you have different dots in the omnivorous graph?
Figure 4: figure fuzzy; labeling of the figures wrong

Experimental design

The idea to calculate the species richness from the gillnet data is great, but I have some concerns about the experimental design:
Why did you use floating gillnets? Why not benthic gillnets? Or other fishing gear? Please discuss that.
In which depth did you set the gillnets? What is the morphology of the sampled lakes? Did they differ in mean depth, maximal depth and size? That might also influence your findings. Please integrate this in your study, if available. Otherwise you should at least clarify that your sampling lakes didn’t differ in these variables.
You compare the different feeding types and migratory/resident species in your manuscript. I recommend testing for differences between benthic and pelagic species as well.
For your data analyses you should use a multivariate correlation to get better results.

Validity of the findings

The findings of the authors show differences in species richness between high-water and low-water season. Moreover the landscape variables seem to influence the species richness. These are good results, but nevertheless the data should be analysed in a more appropriate way, e.g. with a multivariate correlation. The new analyses might lead to better/different results and therefore validity of the findings has to be challenged.

Additional comments

I hope you revise this manuscript in a proper way, because these are great data, that should be published!

---

## Round 0.2 · Minor Revisions

Please complete a thorough edit. Also, please explain in the main text what the measure of fish abundance is (the standardized gill net catch, what is the unit?). Finally, I would like to see the parameters of the models being reported and not only whether things are significant or not.

·

Basic reporting

no comment

Experimental design

no comment

Validity of the findings

no comment

Additional comments

Dear author of the manuscript,
I like this a lot better than the first version. However, there are a few mistakes in grammar etc.

line 87: Herein, we tested......

line 91: (i.e., high- and....

line 161: and 2 m high with varying.....

line 173: time (hours; S1) and the.....

line 326: in this system. Including these..... There are 2 spaces in a row!


Please make sure, that you doublecheck your manuscript again for mistakes like that. Having a manuscript without mistakes supports the validity of your findings!

---

## Round 0.3 · accepted · Accept

You have properly addressed all remaining issues.

#